# Prediction of HAZ Microstructure and Hardness for Q960E Joints Welded by Triple-Wire GMAW Based on Thermal and Numerical Simulation

**DOI:** 10.3390/ma14174898

**Published:** 2021-08-28

**Authors:** Ke Yang, Fei Wang, Dingshan Duan, Bo Xia, Chuanguang Luo, Zhishui Yu, Wang Li, Lijun Yang, Huan Li

**Affiliations:** 1Tianjin Key Laboratory of Advanced Joining Technology, Tianjin University, Tianjin 300072, China; swsyangke@tju.edu.cn (K.Y.); liwang123@tju.edu.cn (W.L.); yljabc@tju.edu.cn (L.Y.); 2School of Materials Engineering, Shanghai University of Engineering Science, Shanghai 201620, China; wangfei@sues.edu.cn (F.W.); Xiabo_sues@163.com (B.X.); yu_zhishui@163.com (Z.Y.); 3Shanghai Collaborative Innovation Center of Laser Advanced Manufacturing Technology, Shanghai 201620, China; 4ROFS Microsystem (Tianjin) Co., Ltd., Tianjin 300462, China; duan_dingshan@163.com; 5Sichuan Institution of Aerospace Systems Engineering, Chengdu 610100, China; chg_luo@163.com

**Keywords:** Q960E, multi-wire welding, SH-CCT, heat affected zone, numerical simulation, thermal simulation

## Abstract

Since heat affected zone (HAZ) is the weak area of welded joints, this article proposes a method to predict the HAZ microstructure and hardness for the triple-wire gas metal arc welding (GMAW) process of Q960E high strength steel. This method combines welding thermal simulation and numerical simulation. The microstructures and hardness of Q960E steel under different cooling rates were obtained by thermal simulation and presented in a simulated HAZ continuous cooling transformation (SH-CCT) diagram. The cooling rate in HAZ were obtained by numerical simulation with ANSYS software for the triple-wire welding of Q960E thick plates. By comparing the cooling rate with the SH-CCT diagram, the microstructure and hardness of the HAZ coarse-grained region were accurately predicted for multiple heat input conditions. Further, an ideal heat input was chosen by checking the prediction results. This prediction method not only helps us to optimize the welding parameters, but also leads to an overall understanding of the process-microstructure-performance for a complex welding process.

## 1. Introduction

As a rising star in the machinery industry, construction machinery has developed rapidly in recent years and has become an important part of the machinery industry [1,2]. Construction machinery includes large excavators, cranes, shovel trucks, forklifts, industrial vehicles and so on. Welding is the main material joining technique for the production of construction machinery [3]. The quality of welding greatly determines the safety of the equipment, and the efficiency of welding is closely related to the processing cost of the equipment [4,5].

At present, there are two major requirements for the welding of construction machinery: First, the welding of high strength steel. Compared with ordinary structural steel, high strength steel offers reduced equipment weight, thereby reducing energy consumption and increasing load [6,7]. The engineering steels have changed from ordinary carbon steels (e.g., Q345 [8]) to high-strength steels (e.g., Q690 [9] or even Q960 [10]). For the welding of high strength steel, a lot of current research has focused on the process windows of different welding methods [11,12,13,14,15], the influence of alloying elements on weldability [16,17,18], the matching and upgrading of welding wire [19,20,21], and the failure of welded joints [22,23,24,25]. Second, the improvement of welding efficiency. Traditional gas metal arc welding (GMAW) process with one wire is increasingly unable to meet the requirement for welding efficiency, while GMAW processes with multiple wires (e.g., tandem GMAW [26,27]) are favored due to its increased welding efficiency. Of course, inter-arc interference may lead to the instability of welding process [28].

Based on the above requirements, we have developed a novel circularly integrated triple-wire pulse GMAW process [29,30]. The three wires are arranged in a triangle in a homemade welding torch, and these wires are insulated from each other and are powered by three pulse welding machines (See details in Section 2.1). We have welded Q235 steel with a deposition efficiency of 15 kg/h, which is almost three times that of single-wire GMAW and is higher than that of tandem GMAW. Besides, the problem of arc interference was solved by alternately generating pulse arcs.

Recently, we have realized the welding of Q960E high-strength steel, and the influence of heat input on the weld joint has been studied [31]. It was found that the coarse-grained region of heat affected zone (HAZ) is the weakest area of the joint. Also, it was necessary to trial many times to find the best process parameters, which makes process optimization very complicated. Therefore, it is very necessary to predict the HAZ microstructure and performance before welding, so as to better optimize the welding process.

In fact, there are currently three main methods for welding microstructure simulation: Monte Carlo method [32,33,34], cellular automata [26,35], and phase field method [36,37]. Monte Carlo method adopts random sampling and kinetic equations to simulate the microstructure evolution. For example, Mishra and DebRoy [34] simulated the grain growth in the HAZ of Ti–6Al–4V welds during TIG welding process; Cellular automata discretize the whole computing area into finite cells, and the state of each cell changes with time and interacts with adjacent packets. Recently, Liu et al. [38] simulated the microstructure evolution in the HAZ of nickel-base alloy weld during TIG welding process; Phase field method is based on the Ginzburg-landau theory and phase field models. Recently, Wang et al. [37] simulated the microstructure evolution of the weld pool for the laser welding of 2A14 aluminum alloy. The above methods are very useful for in-depth understanding of the microstructure evolution of welded joints, but the modelling is difficult, and the calculation is complicated. At present, the research mainly focuses on a few binary or ternary metals.

Aiming at the welding of 960E steel, this article proposes a method for predicting the HAZ microstructures and hardness for the triple-wire pulse GMAW process. This method combines welding numerical simulation and thermal simulation. In the thermal simulation, the HAZ microstructure and hardness under different cooling rates were obtained by using a Gleeble thermal simulator, and the results were presented in a simulated HAZ continuous cooling transformation (SH-CCT) diagram. In the numerical simulation, the cooling rate in HAZ was obtained by using ANSYS software for a certain welding condition (e.g., board thickness and process parameters). The combination of the thermal and numerical simulation can realize a pre-welding prediction for the HAZ microstructure and hardness. The present method is essentially different from the above-mentioned methods because it predicts the phase transition depending on experiment (thermal simulation) rather than tedious and complicated calculation, thus realizing a fast and accurate prediction for the complex welding process (triple-wire welding) and the multi-alloy material (high-strength steel).

It should also be noted that although there are many reports on the thermal simulation for alloy steels, such as Q390 [39], Q690D [40], 3Cr-1Mo-0.25V [41], HSLA [42], X80 [43], X90 [44], and X100 [45], there are almost no reports on Q960E steel. There are many researches on the simulation of welding temperature field for some conventional welding methods, such as TIG welding [46,47], single-wire GMAW [48], submerged arc welding [49], and laser welding [50], but there is no report on the triple-wire welding since it is newly invented.

This article is organized as follows. In Section 2, we introduce the integrated circular triple-wire GMAW, and present the pre-welding prediction method. In Section 3, we show the results of thermal simulation and numerical simulation and verify the correctness of the prediction method by comparing with experimental results. Finally, we summarize this article in Section 4.

## 2. Materials and Methods

### 2.1. Integrated Circular Triple-Wire Pulse GMAW

Figure 1 shows a schematic diagram of the integrated circular triple-wire pulse GMAW system. The core components of the welding system are a homemade welding torch and three Aotai Pulse MIG-500 welding machines (Aotai Electric Co., LTD, Jinan, Shandong, China). Three welding wires were insulated from each other and circularly arranged with an equilateral triangle configuration in the welding torch. The three wires were powered and fed by the three welding machines, respectively. The welding wire close to the front end of the molten pool was called a guide wire, and its corresponding welding machine was called the main welding machine. The two wires located at the rear end of the molten pool were called trailing wire 1 and 2, and their corresponding welding machines were called the slave welding machine 1 and 2. In the welding process, the welding torch was fixed, and the workpiece was moving, which facilitates the shooting of welding arc.

Figure 2a presents the welding arc, welding current and arc voltage for a typical triple-wire pulse GMAW process. In the welding process, bright pulsed arcs alternately appeared on the three welding wires, which can effectively minimize the occurrence of arc breaking [31]. The alternation of the pulse arcs was controlled by the main welding machine, which sends trigger signals to the slave welding machines and itself. This process has been successfully applied to the welding of Q235 steel [29,30] and Q960E steel [31]. A typical weld appearance is shown in the Figure 2b.

### 2.2. Method of Predicting HAZ Microstructure and Microhardness

Figure 3 schematically shows the method for predicting the HAZ microstructure and hardness. This method combines welding thermal simulation and numerical simulation. The SH-CCT diagram, which contains the information of the HAZ microstructure and performance under different cooling rates, can be obtained by thermal simulation. The thermal cycle curve in HAZ, which can reflect the cooling rate, can be obtained by numerical simulation. By comparing the cooling rate with the SH-CCT diagram, we can predict the HAZ microstructure and microhardness for a certain welding condition. Further, we can optimize the welding process by constantly adjusting welding parameters to obtain ideal microstructure and microhardness.

#### 2.2.1. Thermal Simulation of Q960E High-Strength Steel

In this study, Q960E high-strength steel (Wuyang Iron and Steel Co. Ltd., Pingtingshan, Henan, China) was used as the base metal. In delivery state, the Q960E steel is quenched and tempered, and its microstructure is tempered sorbite (See Figure 4). Its chemical compositions and the mechanical properties are shown in Table 1 and Table 2, respectively.

The thermal simulation was performed on a Gleeble3500 thermal simulator (Data Sciences International, INC., St. Paul, MN, USA). As shown in Figure 5a, a sample (Φ6 mm × 90 mm) was fixed by a copper fixture, and a pair of K-type thermocouples was welded on the sample to measure its temperature change. Under the action of resistance heat, the sample can be heated according to a set temperature curve. The expansion process of the sample was recorded by a dilatometer to determine the phase change temperatures. Since steel is a phase-transformed alloy, when it is heated or cooled, in addition to the volume change caused by thermal expansion and contraction, there is also a volume change caused by the phase change. The turning points of the expansion curve will indicate the initial temperature *T*_s_ and the end temperature *T*_f_ of the phase transformation, as shown in Figure 5b.

The phase transformation in steels mainly depends on the chemical composition of the metal and cooling conditions. For the same material and process conditions, the microstructure is mainly affected by the cooling rate. Hence, we used the same heat rate, peak temperature and holding time, but different cooling rates, as shown in Table 3. The cooling rate from 800 to 500 °C is also represented by *T*_8/5_—the cooling time (800 to 500 °C).

After thermal simulation, the microstructures of the samples were observed by an OLYMPUS DP70 optical microscopy (Olympus Corporation, Tokyo, Japan) and the microhardness values were measured by a Huayin HV-1000A micro Vickers hardness tester (Huayin Test Instruments Co., Ltd., Laizhou, Shandong, China).

Take temperature as the ordinate and the time as the abscissa. Multiple temperature-time logarithmic curves under different cooling rates were drawn in the same graph, and then the phase transition points were connected. The curves were also corrected by considering the metallographic microstructure and hardness. An SH-CCT diagram was finally obtained.

#### 2.2.2. Numerical Simulation of the Triple-Wire GMAW Process

The temperature field and thermal history in HAZ can be obtained by numerical simulation. The simulation was performed under the ANSYS 20.0 software platform (ANSYS Inc., Canonsburg, PA, USA). The modelling procedure includes the steps: the establishment of geometric model, meshing, the definition of material properties, heat source loading and boundary condition setting, and solution. Take the butt welding of two 12 mm thickness plates as an example.

Geometric model

The geometric model is shown in Figure 6a. Since the plate was too thick, a groove of 45°, a 1 mm blunt edge and a 1 mm gap were designed. Single-wire GMAW was first used for backing welding, and then the three-wire GMAW was used to complete the main welding. The backing welding adopted a current of 185 A, a voltage of 22.5 V, and a welding speed of 6 mm/s. The three-wire welding adopted a current of 120 A and a voltage of 22.0 V for each wire. Three layers were performed with a welding speed of 8 mm/s. Before welding, the workpiece should be preheated to 100 °C to prevent cold crack.

Meshing

When meshing, the solid 70 element was used in the weld and the area far away from the weld, and the solid 90 element is used for free transition in the other part.

Material properties

The material density was set as 7800 kg/m^3^. The temperature dependent thermal conductivity and specific heat of low carbon steels [51] were used for all calculations, as shown in Figure 7. According to Goldar et al.’s suggestion [51], in the liquid range (>1480 °C) a thermal conductivity of 120 W/(m·°C) was assumed, in order to simulate to a first approximation of the heat transfer by convective stirring in the molten pool. A heat of fusion of 2.1 × 10^9^ J/m^3^ and a heat of transformation of 5.5 × 10^7^ J/m^3^ were associated with the melting (fusion) and transformation temperatures, respectively.

Heat sources and boundary conditions

For the single-wire GMAW, a classic double ellipsoid heat source proposed by Goldar et al. [51] was used in numerical simulation. As shown in Figure 6b, the double ellipsoid heat source is composed of two semi-ellipsoids. The size and shape of the model can be easily adjusted, and the asymmetry situation caused by welding movement can be handled.

According to the Fourier law of heat transfer and the law of conservation of energy, the governing equation for heat transfer of weldments can be established [52]:(1)$$\rho c\frac{\partial T}{\partial t}=\frac{\partial }{\partial x}\left(\lambda \frac{\partial T}{\partial x}\right)+\frac{\partial }{\partial y}\left(\lambda \frac{\partial T}{\partial y}\right)+\frac{\partial }{\partial z}\left(\lambda \frac{\partial T}{\partial z}\right)+q\left(x,\;y,\;z,\;t\right)$$
where *ρ* is the density of the solid, *c* is its specific heat, *λ* is its thermal conductivity and *q* is the power density of the heat source.

For the front quadrant of the heat source, the power density distribution inside the ellipsoid is
(2)$$q(x,\;y,\;z,\;t)=\frac{6\sqrt{3}f_{f}Q}{abc\pi \sqrt{\pi }}e^{{-3x}^{2}/a^{2}}e^{{-3y}^{2}/b^{2}}e^{-3{\left[z+v(\tau -t)\right]}^{2}/c_{1}^{2}}$$

Similarly, for the rear quadrant of the heat source, it becomes:(3)$$q(x,\;y,\;z,\;t)=\frac{6\sqrt{3}f_{f}Q}{abc\pi \sqrt{\pi }}e^{{-3x}^{2}/a^{2}}e^{{-3y}^{2}/b^{2}}e^{-3{\left[z+v(\tau -t)\right]}^{2}/c_{2}^{2}}$$
where *a*, *b*, *c*_1_, *c*_2_ are the ellipsoid width, depth, front edge and back edge, respectively. *Q* is the total energy applied by the heat source, with *Q* = *η* × *U* × *I*. *U* is the welding voltage, *I* is the welding current, and *η* is welding thermal efficiency (*η* = 80%). *f*_f_ and *f*_r_ are the heat distribution parameters for the front and rear ellipsoids, respectively, with *f*_f_ + *f*_r_ = 2. Values of *f*_f_ = 0.6 and *f*_r_ = 1.4 were found to provide the best modelling results [51].

For the three-wire pulse GMAW, we used a composite heat source composed of three double ellipsoids, as shown in Figure 6c. In order to reduce the amount of calculation, the heat source parameters did not change with time, which means an average current was used to replace the pulse current.

For the boundary conditions, we set the areas where the workpiece is in contact with air as convective heat transfer condition and the convective heat transfer coefficient is 20 W/(m^2^·°C). In addition, due to the symmetry, only half of the model on one side of the *yz* plane needs to be established and the *yz* plane (Figure 6c) was set as an adiabatic boundary.

Solving

When solving, the continuous welding process was divided into many small steps. After welding one layer, cool for a certain period to a temperature of about 100 °C. In the cooling process, the time step can be increased to reduce the calculation time.

In order to verify the accuracy of the simulation results, we compared the simulation results with the experimental results. The cross-section of the weld simulated was compared with the actual one. Also, the thermal cycles were measured for several points near the weld, as shown in Figure 6d. The distance *L* between the measuring point and the center of the weld is 9 mm, 10 mm, 11 mm, 12 mm, and 13 mm, respectively.

## 3. Results and Discussion

### 3.1. Welding Thermal Simulation

#### 3.1.1. SH-CCT Diagram of Q960E Steel

Figure 8 shows the SH-CCT diagram of the Q960E steel. The microstructures include lath martensite (ML), lath bainite (BL), and granular bainite (GB). When the cooling rate was less than 2 °C/s, the microstructure of the HAZ coarse-grained region was mainly GB. When the speed was 5–10 °C/s, the microstructure was mainly dominated by BL. When the cooling rate was 10–30 °C/s, the microstructure consisted of BL and ML. When the cooling rate was greater than 50 °C/s, the microstructure was all ML. The hardness values after thermal simulation were also given in the Figure 8. As the cooling rate increased, the hardness increased.

#### 3.1.2. Effect of Cooling Rate on the Microstructure of Q960E Steel

Figure 9 shows the microstructures of Q960E steel after thermal simulation for different cooling rates. When the cooling rate was 0.5 °C/s or 1 °C/s, the microstructure was GB. When the cooling rate was 2 °C/s, the microstructure transformed from GB to BL. As the cooling rate increased to 5 °C/s, the BL was getting finer. When the cooling rate reached 10 °C/s, the ML began to appear, and the microstructure was a mixture of BL and ML. As the cooling rate increased, the content of ML gradually increased. When the cooling rate was greater than 50 °C/s, the supercooled austenite in the steel was almost completely transformed into ML. These results are consistent with the SH-CCT diagram.

### 3.2. Welding Numerical Simulation

Figure 10 shows the temperature fields for the backing welding and the triple-wire filling welding. Due to the symmetry of the temperature field, only half of the workpiece are shown. As shown in the figure, the welding heat source reached the end of the weld bead. The weld was represented by the red area, the temperature of which is above 1480 °C. For the three-wire welding, although the workpiece was heated by three arcs, there was only one weld pool. This is consistent with our experimental observation of the molten pool. Further, the weld morphology can be obtained by slicing the workpiece from the largest section of the molten pool, as shown in Figure 11.

In order to verify the simulation results, we compared the weld morphology simulated with the actual one in Figure 12a. The weld width calculated, and the actual value were, respectively, 8.4 mm and 8.8 mm, and the HAZ width calculated and the actual value were, respectively, 1.8 mm and 1.7 mm. Also, we compared the highest temperatures for the points near the weld in Figure 12b. It can be seen that except for the point closest to the weld (*L* = 9 mm), the error between measurement and simulation was no more than 10% for the other points. Overall, the numerical simulations yielded very accurate results. The source of the error may come from the simplified treatment of the flow of the molten pool and the pulse action of the arc.

It should be noted, however, that since the simulation results cannot be 100% consistent with the actual results, the HAZ position simulated is not always consistent with the actual position. For example, in this case the position of the HAZ coarse crystal region calculated is 8.5 mm from the center of the weld, while the actual position is 9 mm from the center of the weld. Nevertheless, the value of *T*_8/5_ predicted for the HAZ coarse crystal zone is still accurate enough. As shown in Figure 12c, the *T*_8/5_ simulated is 6.1 s and the measured value is 5.4 s. They all indicate that the microstructure after welding is ML, which was consistent with the microstructure we actually observed.

### 3.3. Prediction and Verification

As introduced in Section 2.2, the microstructure and hardness of the Q960E steel at different cooling rates can be obtained by thermal simulation. The cooling rate in HAZ can be obtained by numerical simulation. By combining the results of the two, the HAZ microstructure and performance can be predicted.

Previously, we have conducted a large number of process study for three-wire welding, especially the effect of welding heat input on the microstructure and properties of the weld joint (See [31]). In order to illustrate the accuracy of this method, we predicted the HAZ microstructures and hardness under different heat input conditions and compared with our previous experimental results.

In the experiment, there were four heat input values of 0.99, 1.26, 1.56, and 1.89 kJ/mm, as shown in Table 4. The heat input *E* infers to the welding energy to the weld per unit length. It can be expressed as *E* = *n* ∗ *I* ∗ *U*/*v*, where *n* is the number of welding wires (*n* = 3 in this case). For example, for the test number 1, the heat input is 3 × 120 A × 22 V/10^3^/8 mm/s = 0.99 kJ/mm. The other welding parameters, such as the thickness, groove and gap, were consistent with those described in Section 2.2.2.

Figure 13 shows the microstructures of the HAZ coarse-grained zone under different heat input conditions. When the heat input was 0.99 kJ/mm, the microstructure was mainly dominated by ML and the grains were relatively small. When the heat input was 1.26 kJ/mm, besides ML there were some BL. When the heat input was 1.56 kJ/mm, it was a mixed microstructure of ML and BL, and the grain size increased significantly. When the heat input increased to 1.89 kJ/mm, the microstructure was mainly composed of BL [31].

Table 4 summarizes the microstructure and hardness of the HAZ coarse-grained zones obtained by simulation and experiment. The values of *T*_8/5_ under different heat input values were first obtained by numerical simulation. Then, the hardness and microstructure can be obtained according to the SH-CCT diagram of the Q960E steel. The comparison shows that under different heat input conditions, the HAZ microstructure predicted were completely consistent with the experimental results, and the HAZ hardness predicted is close to the actual value (the error is less than 15%). In general, the prediction was relatively accurate.

Furthermore, ideal welding heat input value can be chosen according to the predicted microstructure and hardness. For construction machinery, there are two important mechanical performance indicators: impact toughness and tensile strength. Impact toughness is related to microstructure. According to the principle of welding metallurgy, the microstructure with the best toughness is a mixture of ML and BL, followed by a single BL, and then a single ML. Strength is related to hardness. Generally speaking, the greater the hardness, the higher the strength. Considering both, we though that the heat input values of tests 2 and 3 are more appropriate. This conclusion was consistent with our mechanical test results of welded joints in [31].

## 4. Conclusions

Taking the triple-wire pulse welding of Q960E high-strength steel as example, this article proposes a method for predicting the HAZ microstructure and hardness by combining the welding thermal simulation and numerical calculation. The following conclusions are drawn:The SH-CCT diagram of Q960E steel, which contains the information of the microstructure and microhardness under different cooling rates, was obtained by thermal simulation.The cooling rate in HAZ was obtained by the numerical simulation of the triple-wire welding process with a three double-ellipsoid composite heat source. Comparing with the test on the weld morphology, the highest temperature near the weld, and *T*_8/5_, it is found that the numerical simulation accuracy was very high.The HAZ microstructure and hardness was predicted by comparing the cooling rate with the SH-CCT diagram. The prediction results were very close to the experimental results, indicating that the prediction method is effective.The welding process can be optimized by adjusting the welding parameters and checking the prediction results.

The prediction accuracy can be further improved by considering the flow of molten pool and the pulse action of the arc. Besides, although only the triple-wire welding of Q960E steel was predicted, this method can be applied to other welding methods and welding materials. These improvements will be done in our forthcoming work.

## Figures and Tables

**Figure 1 materials-14-04898-f001:**
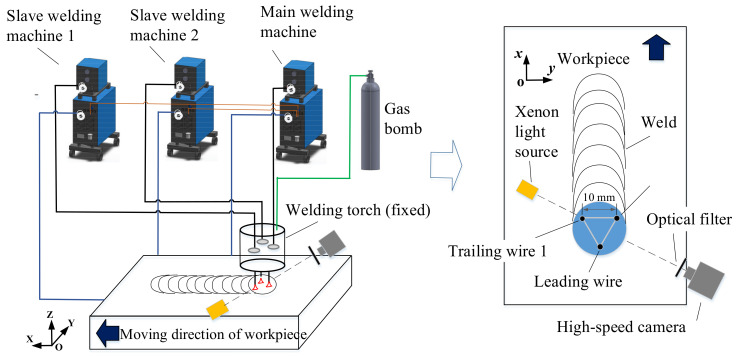
Schematic diagram of an integrated circular triple-wire pulse GMAW system.

**Figure 2 materials-14-04898-f002:**
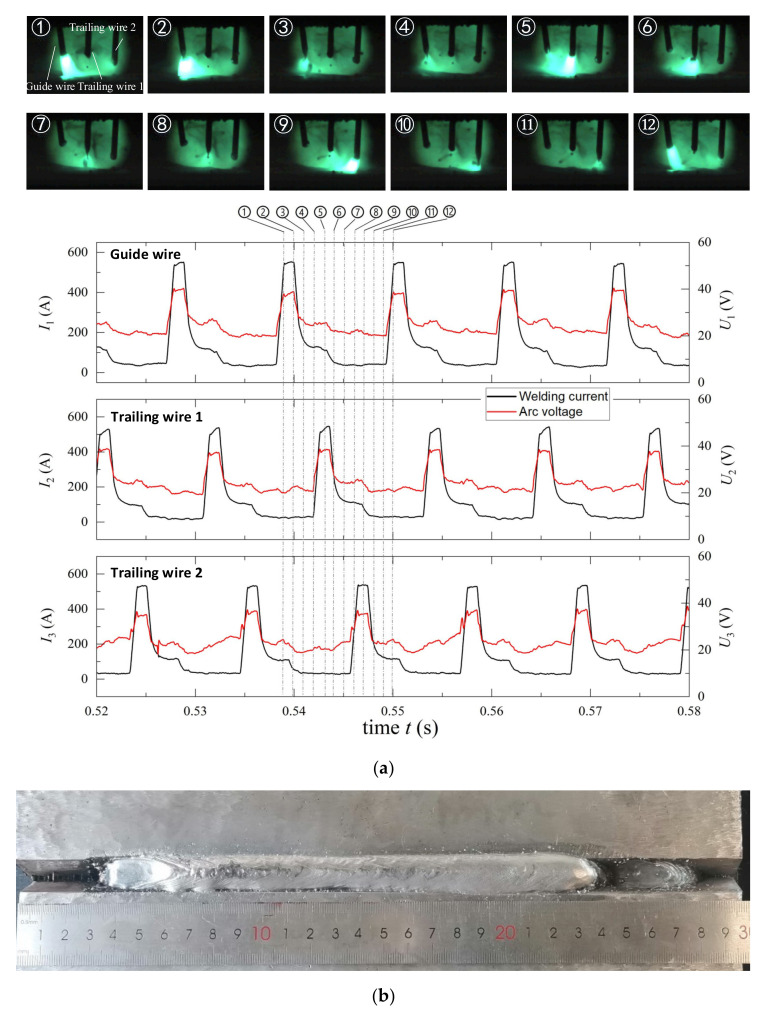
A typical integrated circular triple-wire pulse GMAW process: (**a**) welding arc and its corresponding current and voltage; (**b**) weld appearance.

**Figure 3 materials-14-04898-f003:**
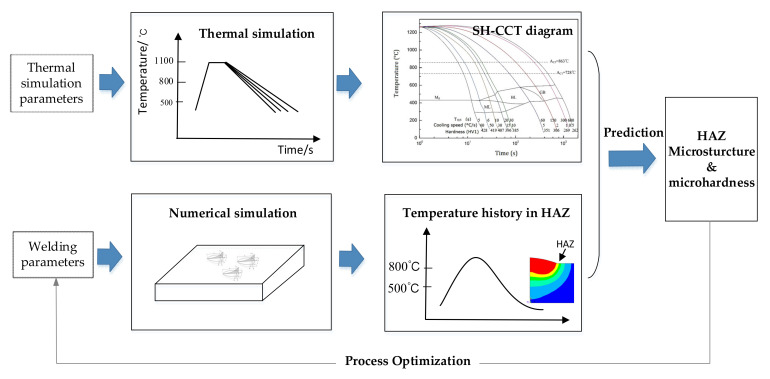
Schematic diagram of the prediction of the HAZ microstructure and microhardness for Q960E joints welded by the triple-wire GMAW process.

**Figure 4 materials-14-04898-f004:**
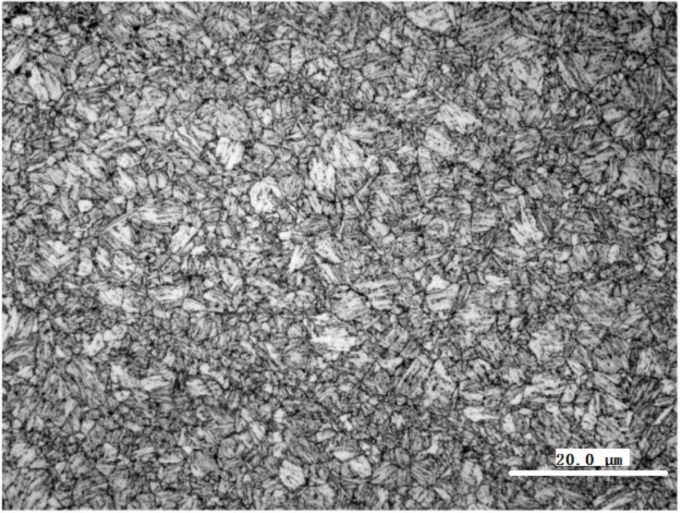
Microstructure of Q960E steel.

**Figure 5 materials-14-04898-f005:**
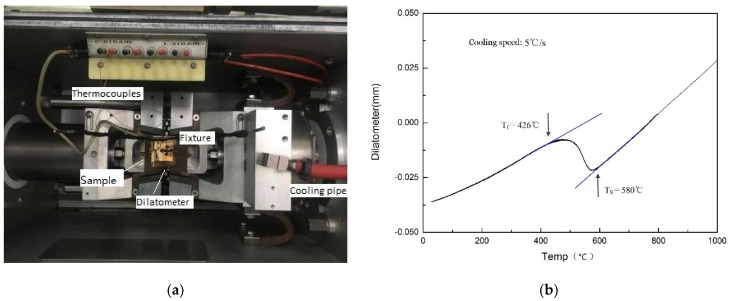
Welding thermal simulation: (**a**) the Gleeble 3500 thermal simulator; (**b**) a typical expansion curve (cooling rate = 5 °C/s).

**Figure 6 materials-14-04898-f006:**
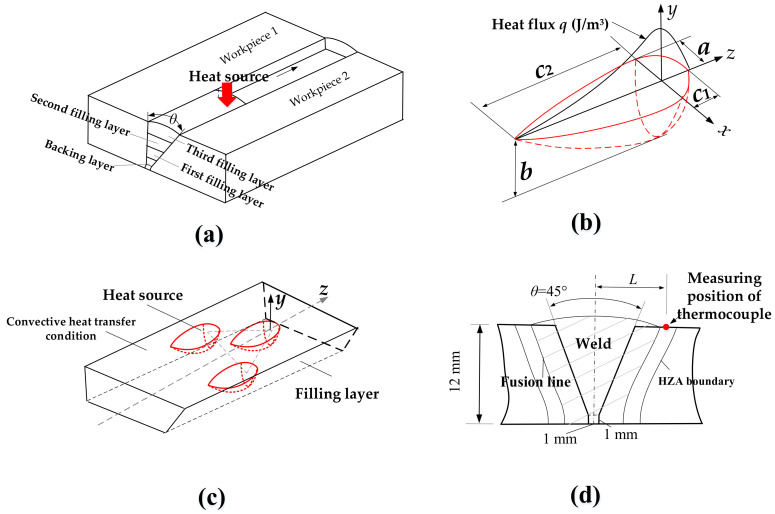
Numerical simulation: (**a**) the geometric model, (**b**) the classic double ellipsoidal model, (**c**) the composite heat source model, and (**d**) the experimental verification.

**Figure 7 materials-14-04898-f007:**
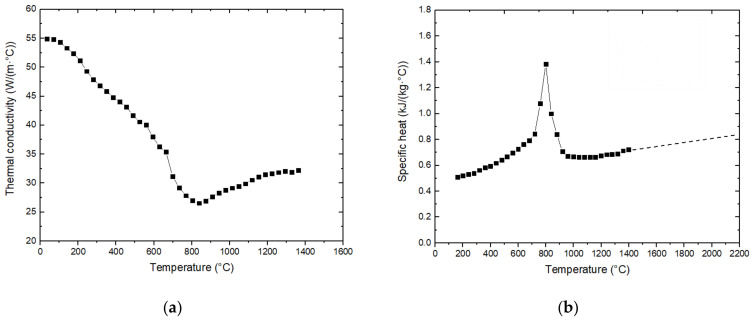
Material properties: (**a**) thermal conductivity; (**b**) specific heat.

**Figure 8 materials-14-04898-f008:**
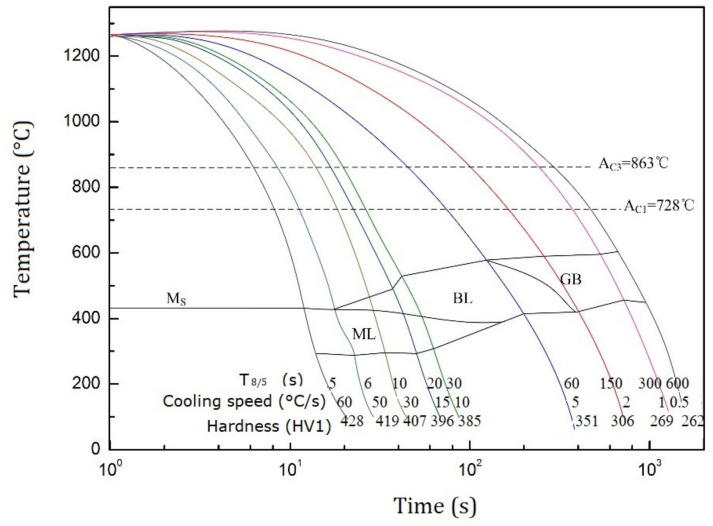
SH-CCT diagram of Q960E steel.

**Figure 9 materials-14-04898-f009:**
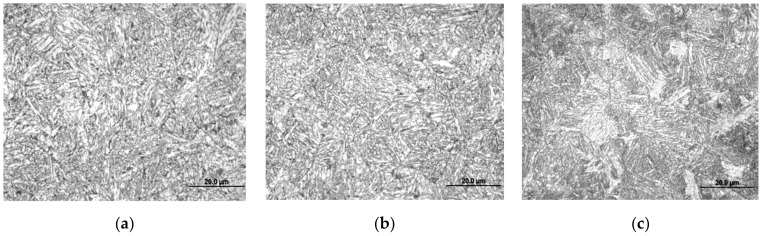
Microstructures of Q960E steel at different cooling rates. (**a**) 0.5 °C/s, (**b**) 1 °C/s, (**c**) 2 °C/s, (**d**) 5 °C/s, (**e**) 10 °C/s, (**f**) 15 °C/s, (**g**) 30 °C/s, (**h**) 50 °C/s, (**i**) 60 °C/s.

**Figure 10 materials-14-04898-f010:**
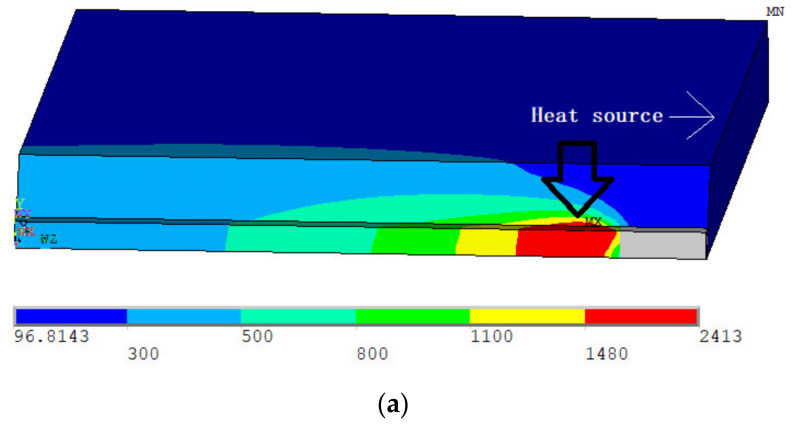
Calculated temperature fields: (**a**) backing welding, (**b**) first layer welding, (**c**) second layer welding, and (**d**) third layer welding.

**Figure 11 materials-14-04898-f011:**
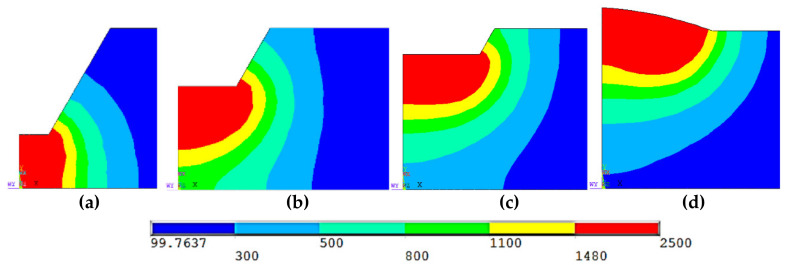
Weld morphology: (**a**) backing welding, (**b**) first layer welding, (**c**) second layer welding, and (**d**) third layer welding.

**Figure 12 materials-14-04898-f012:**
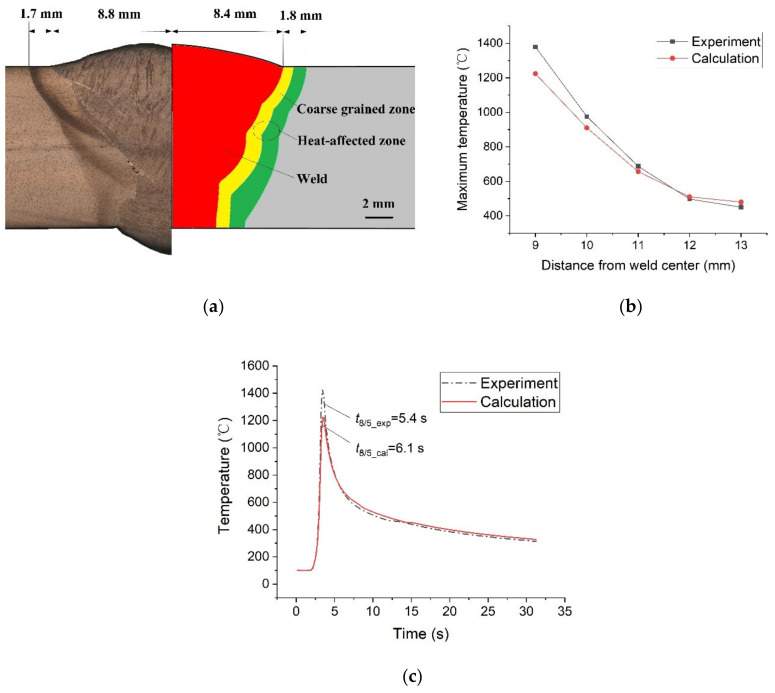
Comparison of simulated results and experimental results: (**a**) weld morphology, (**b**) highest temperatures, (**c**) thermal curve.

**Figure 13 materials-14-04898-f013:**
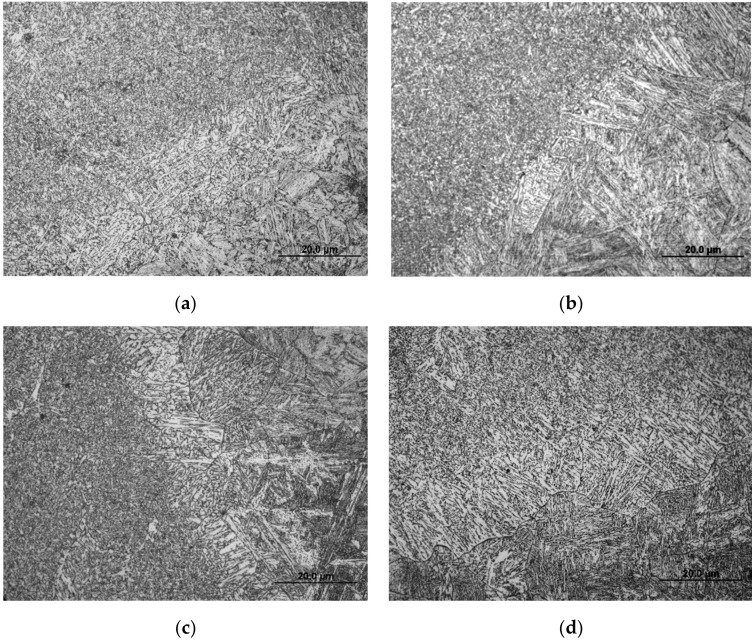
HAZ microstructures under different heat input values [31]. (**a**) *E* = 0.99 kJ/mm (**b**) *E* = 1.26 kJ/mm, (**c**) *E* = 1.56 kJ/mm (**d**) *E* = 1.89 kJ/mm.

**Table 1 materials-14-04898-t001:** Chemical compositions of the Q960E steel and THQ80-1 wire (wt %).

C	Mn	Si	P	S	Cr	Ni	Mo	Ti	Cu	Nb	Ni	Fe
0.16	1.24	0.31	0.014	0.003	0.39	0.61	0.47	0.021	0.02	0.019	0.61	Bal.

**Table 2 materials-14-04898-t002:** Mechanical properties of the Q960E steel.

Tensile Strength *R*_m_ (MPa)	Yield Strength *R* p (MPa)	Elongation *A* (%)	Impact Toughness *A*_kv_ (−40 °C) (J)
1041	1015	14	87

**Table 3 materials-14-04898-t003:** Thermal simulation parameters.

	Heating Rate (°C/s)	Peak Temperature (°C)	Holding Time (s)	*T*_8/5_ (s)	Cooling Rate (°C/s)
1	120	1320	1	600	0.5
2				300	1
3				150	2
4				60	5
5				30	10
6				20	15
7				10	30
8				6	50
9				5	60

**Table 4 materials-14-04898-t004:** Comparation of microstructures and hardness obtained by simulation and experiment.

Test Number	Heat Input E/(kJ/mm)	*T*_8/5_ Calculated/s	Hardness		Microstructure	
			Calculation	Experiment	Calculation	Experiment
1	0.99	6.1	419	422	ML	ML
2	1.26	16.4	399	401	ML+BL	ML+BL
3	1.56	24.5	392	396	ML+BL	ML+BL
4	1.89	36.2	375	380	BL	BL

## Data Availability

The data presented in this study are available on request from the corresponding author.

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
