# Peer review of "Prediction of HAZ Microstructure and Hardness for Q960E Joints Welded by Triple-Wire GMAW Based on Thermal and Numerical Simulation"

_materials, 2021, doi:10.3390/ma14174898_

Round 1
Reviewer 1 Report
Dear authors,
thank you very much for the given insights on modelling and welding with triple-wire pulse welding. The paper is of great interest to the welding community and is well presented. Especially, the combination of welding experiments with moddeling gives a beneficial value. But overall the introduction and the literature you have presented is, for such highly research steel groops way too short. Please overwork the introduction and point out which chalannges needs to be overcome when welding high strength steels, like hydrogen induced cold cracking (f.e. the working group aroaund Kannengießer and Rhode https://doi.org/10.1016/j.ijhydene.2020.05.077)
and the aim for more high strength filler materials (Gehling et al. https://doi.org/10.1007/s40194-021-01086-3)
and the possible usage of undermatching fillers (working group Enzinger https://doi.org/10.1007/s40194-018-0570-1)
and other hot topics for the group of high strength steels.
Please also include a discussion section in which you relate the presented work to the state of the art and to the approach of other scientists in simulating microstructure evolution and point out particularities. This should also include the achieved microstructure morphology. This chapter should also address the need for further investigation of your process in terms of burning of the alloying elements, hydrogen input welding fumes and others. Pleas add here additional literature. I estimate that you easily find another 20 sources wicht are supporting your work and which can be used to point out what is new in your work.
best regards
a reviewer
Author Response
On behalf of all the contributing authors, I would like to express our sincere appreciations of your constructive comments concerning our article. These comments are valuable and helpful for improving our article. Based on them, we have made careful modifications on the original draft. We hope the new manuscript will meet your journal's standard. You will see our point-by-point responses to the reviewer' comments/questions in the attachment.

Reviewer 2 Report
An interesting process is presented
But the performance needs improvement
no equations are presented in section "3.2. Welding numerical simulation". The equations describing the model, the main boundary conditions, etc. must be presented
Author Response
On behalf of all the contributing authors, I would like to express our sincere appreciations of your constructive comments concerning our article. These comments are valuable and helpful for improving our article. Based on them, we have made careful modifications on the original draft.
Below you will see our point-by-point responses to the reviewer' comments/questions in the attachment.

Reviewer 3 Report
The paper presents an analysis of the heat-affected zone, microstructure and resulting tribological properties of a gas metal arc weld on steel. The paper is generally well written and can be accepted for publication subject to the following revisions.
- Try to avoid abbreviations in the title where possible, although it could be challenging in this case.
- Introduce full forms when abbreviations appear first. For example, GMAW in the second line of the abstract.
- Present all the material properties used for the finite element model in a table. Adding on a detailed modelling procedure will also be useful.
- What are the limitations of the numerical model presented?
Author Response

(The authors gave the same response as above.)

Round 2
Reviewer 1 Report
Dear authors,
thank you very much for the made changes,
i can agree on your comments and don't have additional comments for you.
Best regards
a Reviewer